# *In Vitro* Propagation of Three Date Palm (*Phoenix dactylifera* L.) Varieties Using Immature Female Inflorescences

**DOI:** 10.3390/plants12030644

**Published:** 2023-02-01

**Authors:** Ahmed M. Abdelghaffar, Said. S. Soliman, Tarek A. Ismail, Ahmed M. Alzohairy, Arafat Abdel Hamed Abdel Latef, Khadiga Alharbi, Jameel M. Al-Khayri, Nada Ibrahim M. Aljuwayzi, Diaa Abd El-Moneim, Abdallah. A. Hassanin

**Affiliations:** 1Genetics Department, Faculty of Agriculture, Zagazig University, Zagazig 44511, Egypt; 2Department of Botany and Microbiology, Faculty of Science, South Valley University, Qena 83523, Egypt; 3Department of Biology, College of Science, Princess Nourah bint Abdulrahman University, Riyadh 11671, Saudi Arabia; 4Department of Agricultural Biotechnology, College of Agriculture and Food Sciences, King Faisal University, Al-Ahsa 31982, Saudi Arabia; 5Department of Plant Production, (Genetic Branch), Faculty of Environmental and Agricultural Sciences, Arish University, El-Arish 45511, Egypt

**Keywords:** date palm, growth regulators, heritability, immature female inflorescence, tissue culture

## Abstract

Immature female inflorescences are promising materials for use as explants for the tissue culture of date palm. Four types of MS media were used in this study during the four micropropagation stages—starting media (SM), maturation media (MM), multiplication media (PM) and rooting media (RM)—to micropropagate three elite date palm varieties, Amri, Magdoul and Barhy using the immature female inflorescences as explant. The highest percentage of callus induction in all the varieties studied was obtained on the SM1 (9 µM 2,4-D + 5.7 µM IAA + 10 µM NAA). Culturing on the MM1 (4.5 µM 2,4-D + 9.8 µM 2-iP + 1.5 AC) allowed us to obtain the best value in terms of callus weight. After culturing on the PM1 (4.4 µM BA + 9.8 µM 2-iP) produced the highest numbers of somatic embryos and shoots. The explants on RM2 (0.5 µM NAA + 1.25 µM IBA + 3 g AC) showed the highest root numbers and root lengths, while the highest shoot length was achieved on RM3 (0.5 µM NAA + 0.5 µM IBA + 3 g AC). The Amri variety presented the best response among the three varieties in all parameters, followed by the Magdoul and Barhy varieties. In all the stages of micropropagation, the analysis of variance revealed highly significant variations among varieties and culture media, and a significant difference in the number of roots during the rooting stage. The results also showed non-significant differences in the interaction between varieties and culture media, except for shoot length in the rooting stage. The results also reveal the broad sense heritability ranging from low to high for the measured parameters. It can be concluded that the immature female inflorescences can be used as a productive explant source for successful date palm micropropagation using the SM1, MM1, PM1 and RM2 culture media. It can also be concluded that the success of date palm micropropagation not only depends on the concentrations of growth regulators, but also on their types.

## 1. Introduction

The date palm (*Phoenix dactylifera* L.) is a diploid (2*n* = 36), perennial, long-lived monocot plant that belongs to the Arecaceae family and has a dioecious nature, which is highly heterozygous [1]. Due to its economic and dietary significance, it is the most significant fruit crop grown under warm climate conditions, especially in North Africa and the Middle East [2]. It is traditionally propagated for a long time by offshooting, which is inefficient because each tree only produces a few offshoots, especially in some elite varieties, and fruit-bearing can take up to seven years [3]. There are even cases in which no offshoots are produced at all [4]. Tissue culture propagation is the best way to produce high-quality and efficient plants in date palm [5]. Useful explants for propagation in commercial labs around the world are shoot tips; however, the major downside here is that the entire offshoot is sacrificed [6]. Nutrient medium and the exogenous application of plant growth regulators (PGRs) have a vital role in *in vitro* growth, differentiation, and plant regeneration of date palm [7]. The cytokinin type and concentration are crucial for organogenesis. Thidiazuron and benzyladenine are the most preferably used cytokinins in the regeneration systems in some fruit trees [8]. The auxin 2,4-D has been widely used at high concentration in date palm micropropagation. However, it was reported that this may induce somaclonal variation within regenerants [9]. Using plentiful and reliable sources, such as the immature female inflorescences of date palms, is considered the most promising method for the propagation of elite varieties of female and male date palms [10]. The use of the immature female inflorescences method will shorten the time required for the production of a large number of date palm plants to 1 to 2 years, compared to 3–4 years in the case of using a shoot tip. This will reduce the time and effort needed to generate thousands of plants through embryogenesis, in addition to avoiding several problems encountered with shoot tips, such as the high rates of contamination and browning, and the protracted start phase [11]. The inflorescences were easily converted to a vegetative state; therefore, the use of meristematic floral spikes at an early stage is intended to achieve this [12]. Moreover, these early-stage changes depend on the variety, environment, and state of the mother plant [6]. Additionally, the ability of the floral tissue to reverse into a vegetative state is affected by various factors, such as the age of the immature inflorescences at the time of excision and the components of the initiation medium, particularly the plant growth regulators and the sequence of nutrient media used is decisive [13]. This is related to the proper direction of each morphogenetic type to its subsequent suitable stage of growth and nutrient medium. Moreover, determining the stage of excision can vary by cultivar, climatic conditions and the nutritional state of the mother plant [14]. On the other side, Kriaa, et al. [15] proposed a protocol based on the use of mature female flowers, taken at the latest developmental stage, before the opening of the spathe.

This study presents a new perspective on date palm micropropagation for three important date palm varieties—Amri, Barhy and Magdoul—using immature female inflorescences as explants and using different plant growth regulators to achieve rapid responses of callus induction and shoot and root formation. This study aimed to determine the ideal tissue culture media particularly the concentration and the type of plant growth regulators for successful date palm micropropagation using immature female inflorescences, and to provide a new protocol for the micropropagation of date palm using explants that are abundantly available every year and can, therefore, be used as cheap and potent micropropagation sources without sacrificing adult trees or offshoots.

## 2. Results

### 2.1. Starting Stage

The immature inflorescence explants of the three cultivars (Amri, Magdoul and Barhy) were cultured on MS medium containing different plant growth regulators to study the effects of different combinations and concentrations on the induction of callus and embryogenic calli. After 3–4 weeks of culture, the data recorded on the initial responses of the explants revealed significant variations among the date palm varieties. The analysis of variance for callus induction frequency showed significant differences in the main effects of varieties and media. The interaction effect of these factors was not significant (Table 1). The SM1 medium (9 µM 2,4-D + 5.7 µM IAA + 10 µM NAA) showed the best response by producing the highest callus induction frequency (69.5%), indicating significant differences from the other starting media. SM2 (4.5 µM 2,4-D + 2.85µM I AA + 5 µM NAA) occupied the second position, as it produced good callus induction frequency percentages (54.93%). Meanwhile, the lowest percentage was recorded on the SM3 medium containing 5.7 µM IAA + 5 µM NAA in Amri (43.19%) (Figure 1A). The Amri variety showed the best response by producing the highest callus induction frequency (68.57%), indicating significant differences from the other varieties. The Magdoul variety occupied the second position, as it produced good callus induction frequency percentages (58.94%). Meanwhile, the lowest percentage was recorded with the Barhy variety (40.1%) (Figure 1B). The SM1 medium achieved the shortest period of callus induction (25.162 days) indicating significant difference from both of the SM2 medium (27.66 days) and SM3 medium (30.29 days) (Figure 1C). The Amri variety achieved the shortest period of callus induction (25.16 days) indicating non-significant difference from Magdoul (25.81 days) and significant difference from Barhy (32.14 days) (Figure 1D).

### 2.2. Maturation Stage

The callus weight increased significantly in this stage (Figure 2B, Figure 3B and Figure 4B). The weight of the callus and embryogenic callus was strongly influenced by the concentrations of both 2,4-D and 2-ip during that stage. It is evident from the results in Figure 5A that the maturation medium MM1 produced higher callus weights (1.99 g) than the MM2 medium (1.30 g) and the MM3 medium (0.84 g). These results show that the callus grown on the MM1 medium (4.5 µM 2, 4-D + 9.8 µM 2-iP + 1.5 AC) exhibited a better weight than the MM2 and MM3 media. The Amri variety produced higher callus weights (1.69 g) than the Magdoul variety (1.39 g) and the Barhy variety (1.04 g) (Figure 5B). The ANOVA shows highly significant differences among the three maturation media and the three varieties in terms of callus induction frequency percentage, the number of days to initiate callus and callus weight. No significant differences were recorded in terms of the interaction between media and varieties. The results also reveal high broad sense heritability (82.96% and 81.64%) for callus induction frequency (%) and days required to initiate callus, respectively, as well as low broad sense heritability (47.68%) for callus weight (Table 1).

### 2.3. Multiplication Stage

Three different types of media were used for the multiplication of the calli resulting from the maturation stage to investigate their effects on the numbers of somatic embryos and shoots in the three studied varieties (Figure 2C, Figure 3C and Figure 4C). According to the LSD_0.05_ test, significant differences were recorded between the three varieties on the three types of multiplication media for the number of somatic embryos and the shoots (Figure 6). These results show that the PM1 medium (4.4 µM BA + 9.8 µM 2-iP) showed significant differences from the PM2 and PM3 media for the number of somatic embryos and shoots number with the three varieties. The multiplication medium PM1 produced a higher number of somatic embryos (6.77) and shoots (10.39) than the PM2 medium (4.74) and (7.33) for number of somatic embryos and shoots, respectively. The PM3 medium produced the lowest number of somatic embryos (3) and shoots (5.22) (Figure 6A,C). The Amri variety produced higher number of somatic embryos (6.99) and shoots (10.76) than the Magdoul variety: (4.55) and (7.22) for number of somatic embryos and shoots, respectively. The Barhy variety produced the lowest number of somatic embryos (2.96) and shoots (4.96) (Figure 6B,D). The ANOVA showed highly significant differences among the three types of multiplication media and the three varieties in terms of the number of somatic embryos and shoots. No significant differences were recorded in the interaction between media and varieties. The results also reveal low broad sense heritability: 43.569% and 52.36% for the number of somatic embryos and the shoots number, respectively (Table 2).

### 2.4. Rooting Stage

Shoot rooting happened concurrently with the elongation of the leaves (Figure 2D, Figure 3D and Figure 4D). The results showed significant differences among the four rooting media in terms of the number of roots (Figure 7A), root length (Figure 7A) and shoot length (Figure 7E), according to LSD_0.05_ values. The RM2 medium (3.3 g/L MS + 50 g/L sucrose + 6 g/L agar + 0.5 µMN AA + 1.25 µM IBA + 3g AC) showed the maximum root number and root length, while the lowest root number and root length were recorded with medium RM3 (3.3 g/L MS + 30 g/L sucrose+ 6 g/L agar + 0.5 µM NAA + 0.5 µM IBA + 3g AC). On the other side, The Amri variety presented the best response among the three varieties in terms of root number, root length and shoot length, while Barhy presented the lowest response (Figure 7B,D). The highest shoot length was achieved on the RM3 medium, while the lowest shoot length appeared on the RM2 medium (Figure 7E). The ANOVA showed highly significant differences among the four rooting media in terms of root number, root length and shoot length, and highly significant differences among the three varieties in terms of both root length and shoot length, while the difference was significant in root number. No significant differences were recorded in the interaction between media and varieties except in terms of shoot length. The results also revealed low (22.004), intermediate (68.41) and high (80.11) broad sense heritability for root number, root length and shoot number, respectively (Table 3).

### 2.5. Pre-Acclimatization and Acclimatization Stages

The Amri variety showed the highest growth vigor, followed by Magdoul and Barhy, according to the visual description of Pottino [16]. The percentage of acclimatized plantlets was 86.67% in the Amri variety, followed by Magdoul (82.33%) and Barhy (77.56%) on peat moss and Perlite (2/1 *v/v*) (Figure 2E, Figure 3E and Figure 4E).

## 3. Discussion

The callus induction in date palm micropropagation was significantly influenced by several variables, including variety, the size of the explant, the time of explant excision, the physiological state of mother plant, the culturing period and type, and the concentration of plant growth regulators [6,14,17]. The decisive factors in date palm micropropagation by immature female inflorescence explants are the components of the culture medium particularly the growth hormons and the pace of application of the nutrient media throughout the micropropagation protocol [11]. The current study discusses the differences among varieties and the roles of nutrient media in the responses of callus induction in a short period. The explants’ effects on callus induction frequency increased with the SM1 medium and eventually declined to a low percentage in the SM3 medium. In contrast, the SM3 medium required the shortest period to initiate callus induction. These results reveal that the presence of 2, 4-D and the concentration of NAA promoting the percentage of callus induction frequency may slightly increase the period required for callus induction. This result was proven by Solangi, et al. [18] who obtained the highest percentage of callus induction in the Aseel variety using a medium containing 2 mg/L of NAA, while in the Dhakki variety the highest value was recorded on a medium containing 2 mg/L of 2, 4-D. The lowest value, however, was seen in the explants of both varieties cultured on 0.1 mg/L NAA or 2, 4-D, indicating the critical need of date palm cells for high concentrations of auxins to promote the percentage of callus induction.

The maturation medium MM1 (4.5 µM 2, 4-D + 9.8 µM 2-iP + 1.5 AC) increased the callus weight compared to the MM2 and MM3 maturation media, which included lower amounts of 2, 4-D and 2-iP, indicating the importance of the high quantity of growth regulator not only for callus induction frequency but also for obtaining the best callus weight. Al-Khairi, et al. [19] reported that 2, 4-D can be considered one of the most helpful auxins: it has been utilized in successful cultures as it acts as a growth promoter and enhances the growth of the cells. Additionally, several studies have documented the beneficial effects of various treatments of 2, 4-D in combination with other growth regulators to promote the embryogenesis of calli in various date palm varieties [20,21].

Higher numbers of somatic embryos and shoots were produced in the early differentiated and mature explants kept in dark conditions that were shifted onto multiplication media, PM1 medium (4.4 µM BA + 9.8 µM 2-iP) than explants cultured on PM2 (2.2 µM BA + 4.6 µM kin) and PM3 (0.5 µM NAA + 0.5 µM IBA + 3 g AC),illustrating that multiplication is promoted by higher concentrations of BA and 2-iP growth regulators which explain how the response of date palm explants is affected by both the concentration and the type of cytokinins. The results of the study by Loutfi and Chlyah [22] are consistent with the current results, in which they indicated that the medium with NAA 0.5 mg/1 or BAP 2 mg/L and 2ip 1 mg/L yielded the best shoot multiplication rate. Additionally, Aslam and Khan [23] reported the positive effects of BAP 7.84 μM on the formation of shoots. Interestingly, Taha, et al. [24] discussed the interaction between the strength of the culturing media and growth regulators. They found that the use of full-strength MS media instead of half-strength MS media increased the proliferation rate when using 3 mg/L 2-ip and 0.5 mg/L NAA.

The rooting stage of date palm micropropagation is crucial for the success of date palm tissue culture [19]. In the current research, the root number and root length are gradually promoted in parallel with decreasing the strength of the rooting MS media. The RM2 medium (3.3 g/L MS + 50 g/L sucrose + 6 g/L agar + 0.5 µM NAA + 1.25 µM IBA + 3 g AC) presented the highest root number and root length in spite of possessing the same composition of RM1, aside from the quantity of sucrose. This important result proves the improvement in root formation and development on high-sucrose medium compared to low-sucrose medium. This result is consistent with those obtained by [25]. In contrast to the root number and root length, the highest shoot length was obtained by the RM3 medium (3.3 g/L MS + 30 g/L sucrose+ 6 g/L agar + 0.5µMNAA + 0.5 µM IBA + 3 g AC) containing low concentrations of IBA and sucrose, which confirms the counter effect of auxins on shoot formation and development. Several studies have discussed root formation and development; Alansi, et al. [26] claimed that the root length was enhanced by using ½ MS medium containing 0.25 mg/L of ABA and 45 g/L of sucrose, while increased root number was promoted with 3/4 MS medium strength. Refs. [24,27] compared ¼, ½, ¾ and full MS strength; they found that using ¾ MS salts produced the best results for root production. Metwali, et al. [28] found that NAA at 2.5 or 2.0 mg/L in combination with low concentrations of IBA at 0.0 or 0.5 mg/L were the best concentrations to induce roots and improve the mean lengths of roots/shoots in the Safawi and Magdoul varieties, respectively.

The high broad sense heritability for callus induction frequency (%), the number of days to initiate callus, and the shoot length, along with the intermediate broad sense heritability for root length, illustrate that those parameters could be improved through the selection and improvement of varieties. On the other hand, the low broad sense heritability for callus weight (g), the number of somatic embryos, and the number of shoots and roots indicates that we can promote these traits by modifying the culture media through altering the type and the concentration of the growth regulator in the different culture stages.

The acclimatization of the produced plantlets with a complete root system to greenhouse conditions is revealed to be particularly critical for the successful establishment of field plants [29]. The varied percentages of acclimatized plantlets (86.67% in the Amri variety, 82.33% in the Magdoul variety and 77.56% in the Barhy variety) on perlite and peat moss (2:1 v:v) may be due to fungal attack, which increased as a result of high moisture content. Tisserat [30] reported that the highest rate of survival was found in date palm plantlets of 10–12 cm that were transplanted into a peat moss–vermiculite mixture (1/1: *v/v*) and covered with transparent plastic compared to the shorter plantlets. Bekheet [31] reported that a planting mixture made up of an equal quantity of vermiculite, peat, and sand produced the greatest results, and the percentage of survival was 80% after 18 months. Abahmane [32] suggested that the lack of root cell differentiation caused by a shortage of growth regulators or sugar might be responsible for the death of date palm plantlets during acclimatization.

The improvement of date palm production not only depends on the *in vitro* propagation techniques but also the use of chemical and physical mutagens [33], the recent genetic approaches [34,35], phylogeny analysis [36], and molecular markers [37,38].

## 4. Materials and Methods

### 4.1. Plant Materials

The experiments in this study were performed at the Genetics department, Faculty of Agriculture, Zagazig University, Egypt, in the three consecutive seasons 2019, 2020, and 2021. Immature female inflorescences with 20–30 cm lengths of Amri, Magdoul and Barhy cultivars were obtained from the mother plants in the spring season, covered by plastic film and transferred to the laboratory for further procedures.

### 4.2. Sterilization and Explants Preparation

The spathes were initially cleaned under running water, surface sterilized in 60% Clorox (5.25% sodium hypochlorite) for 30 min and then rinsed in sterilized distilled water for 30 to 60 s under aseptic conditions (Figure 8). The sheath was opened from the middle on one side, and spikelet explants were divided into parts containing 2–3 florets and aseptically cultured on starting media.

### 4.3. Preparation of Culture Media

Nutrient MS medium [39] was used in the current investigation. Full-strength (4.4 g/L) and three-quarter-strength (3.3 g/L) MS medium was used in this experiment; 3% of sucrose was added into the culture medium and the growth regulators were added as mentioned in Table 4 before the adjustment of pH. Before autoclaving, the pH of the medium was adjusted to 5.7 ± 0.1 using 0.1 M HCl and 0.1 M NaOH. Next, 7 g/L agar (PTC agar, sigma) was added. After that, the medium underwent a 20 min autoclave at 121 °C. Laminar flow was employed to pour the sterile medium into 50 mL sterilized screw-top jars and this was allowed to solidify.

### 4.4. Culture Conditions

The spikelets were cut into parts of 3 cm length including 3–4 immature florets and cultured on three different starting media (Table 4) in three replicates, each jar containing 4 explants with 4 florets per explant. All cultured explants were incubated under full darkness in an incubator at 27 °C. Incubated explants were subcultured twice every 21 days on the same starting medium. Explants that responded well were placed in three different maturation media (Table 4) for 1–2 subcultures. All cultures were kept in darkness (24 h) for six weeks with subcultures taken every 21 days on the same media. Matured and early-differentiated explants in the dark were moved to three types of plant multiplication media under light conditions (16 h light and 8 darkness) for 2–3 subcultures for three months. Clusters of developed shoots that resulted from the multiplication stage were transferred to four kinds of 3/4 strength MS rooting media (Table 4) for 24 weeks with 3–4 subcultures. Cultures were incubated at 27 ± 2 °C with a light intensity of 3000 lm/m^2^ for 18 weeks and 4000 for 8 weeks.

### 4.5. Measurements

At each subculture’s initial stage, callus induction frequency, callus weight according to Loyola-Vargas and Vázquez-Flota [40] with some modifications, and callus induction days were recorded. The following formula was used to determine the callus induction frequency:$$\mathrm{Callus}\;\mathrm{induction}\;\mathrm{frequency}=\frac{\mathrm{Number}\;\mathrm{of}\;\mathrm{florets}\;\mathrm{developed}\;\mathrm{to}\;\mathrm{calli}}{\mathrm{Total}\;\mathrm{number}\;\mathrm{of}\;\mathrm{cultured}\;\mathrm{florets}}\times 100$$

The number of somatic embryos/jar and shoot number/jar was estimated after every subculture. After 6 months, the root number, root length (cm) and shoot length were recorded. Heritability in the broad sense was calculated from the following equation [41]:
$$h^{2}=\frac{\sigma_{g}^{2}+\sigma_{i}^{2}}{\sigma_{g}^{2}+\sigma_{i}^{2}+\sigma_{e}^{2}}\times 100$$
where *σ*^2^_*g*_, *σ*^2^_*i*_ and *σ*^2^_*e*_ are the variances due to varieties, interaction and error, respectively.

### 4.6. Pre-Acclimatization

Well-rooted plantlets were shifted to ^1^/_2_ MS (2.2 g/L) medium without hormones supplemented with 30 g sucrose plus 1.5 g activated charcoal (AC) and 2.0 mg/L Ca-pantothenate, and then cultured on ^1^/_2_ MS medium without hormones including 15 g sucrose and 1.5 g AC. Plantlets were kept on this medium until they were a sufficient height, had a healthy root system and had two to three leaves, and then moved to sugar-free liquid medium in tubes covered with aluminum caps to reduce light emission. Plantlets were moved from ordinary small tubes to larger ones. Cultures were maintained in a high-light environment (9000 µmol/m^2^/s). This stage required about 3 months (two subcultures).

### 4.7. Acclimatization

The rooted plantlets were carefully collected from tube, then washed under running water and rinsed with Rezolex fungicide (100 g) solution (Generic) for 10 min. The plantlets were then transplanted into pots filled with a 2:1 *v/v* peat:perlite mixture [42]. Under a cover of white translucent polyethylene sheets, plantlets were housed for 4–6 weeks at 27.2 °C with natural daylight and high relative humidity (90–95%) conditions. The polyethylene sheets were gradually taken off to let the plants become familiar with the greenhouse environment. Plants with 2–3 leaves were irrigated twice weekly for the next 24 weeks with 1/4 MS nutrients. To maintain the relative humidity at 80–90%, plantlets with mainly upright leaves were moved to the soil under a low-transparency plastic tunnel. The percentage of acclimatized plantlets was calculated by dividing the number of plants that successfully adapted to their new environment by the total number of plants.

### 4.8. Statistical Analysis

The current study was set up as a factorial experiment in a completely randomized design comprising two main factors: culture system of three varieties and medium composition at three levels at all stages except rooting stage with four levels of media. Each treatment consisted of three replications. The data are presented as an average ±standard deviation of the replications. Data were subjected to analysis of variance (ANOVA) to locate the significant differences among means squares. When significant differences were found, L.S.D_0.05_ post hoc was used for multiple comparisons among mean values at *p* < 0.05 according to [43] using Statistix 9 software.

## 5. Conclusions

Determining the ideal tissue culture medium, focusing on the type and the concentration of growth regulators and the productive explants, is crucial for the success of date palm in vitro propagation. According to the current results, it could be concluded that the starting medium SM1 (9 µM 2,4-D + 5.7 µM IAA + 10 µM NAA) was the best for the starting stage in all studied varieties. The maturation medium MM1 (4.5 µM 2, 4-D + 9.8 µM 2-iP + 1.5 AC) allowed us to obtain the highest callus weight. The multiplication medium PM1 (4.4 µM BA + 9.8 µM 2-iP) produced the highest number of somatic embryos and shoots. The rooting medium RM2 (0.5 µM NAA + 1.25 µM IBA + 3 g AC) produced the highest root number and root length, while the highest shoot length was obtained by RM3 (0.5 µM NAA + 0.5 µM IBA + 3 g AC). The immature female inflorescence is a promising explant for the micropropagation of date palm as it provides a productive explant without sacrificing the adult trees or the offshoots. We can also conclude that the success of date palm micropropagation not only depends on the concentration of the growth regulators, but also on their types.

## Figures and Tables

**Figure 1 plants-12-00644-f001:**
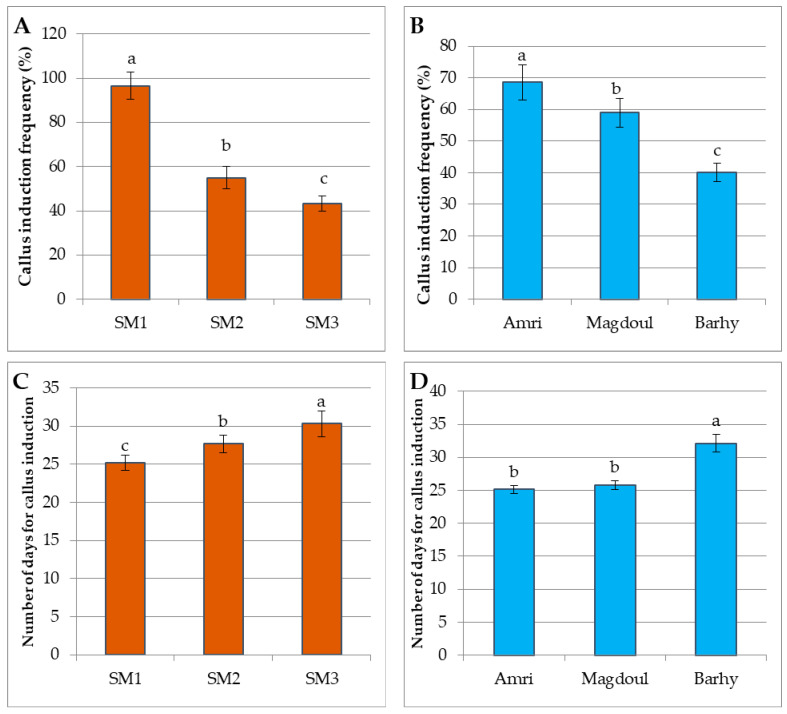
(**A**). The percentages of callus induction frequency in the three varieties (Amri, Magdoul and Barhy). (**B**). The percentages of callus induction frequency of the three starting media (SM1, SM2 and SM3). (**C**). The number of days for callus induction on the three starting media (SM1, SM2 and SM3). (**D**). The number of days for callus induction in the three varieties (Amri, Magdoul and Barhy. Note: Charts with the same letters are not statistically different at *p <* 0.05. Different letters denote a significant difference among culture media at *p <* 0.05.

**Figure 2 plants-12-00644-f002:**
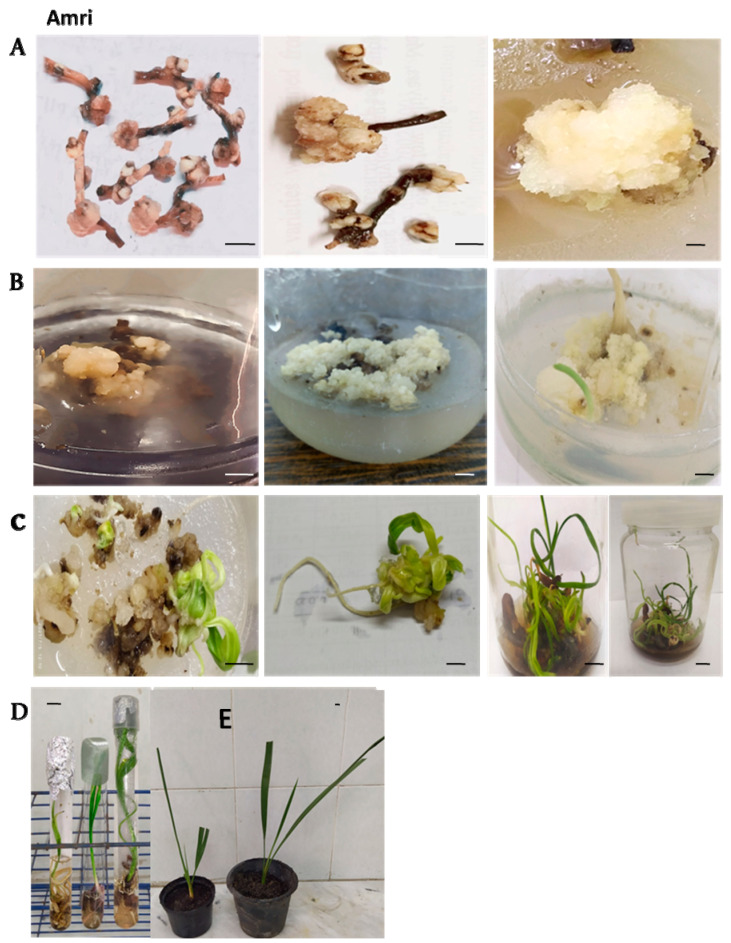
Date palm micropropagation by immature female inflorescences in Amri variety (**A**). Various structures of callus formation and development on starting media after the first subculture, the second subculture and the third subculture, respectively (from left to right). (**B**). Matured embryogenic calli and development of somatic embryogenesis under darkness after 8 weeks, 12 weeks and 16 weeks, respectively (from left to right). (**C**). Germinated somatic embryos and shoot multiplication after 4 weeks, 6 weeks and 12 weeks, respectively (from left to right). (**D**). Shoot rooting. (**E**). Acclimatized plantlets. Scale bar: 1 cm.

**Figure 3 plants-12-00644-f003:**
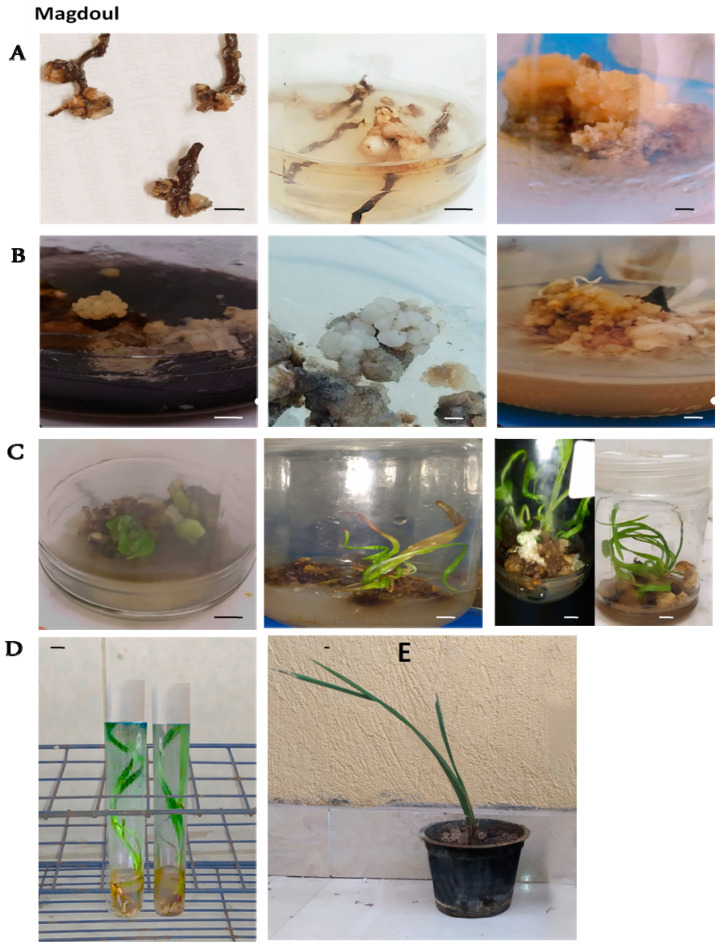
Date palm micropropagation by immature female inflorescences in Magdoul variety. (**A**). Various structures of callus formation and development on starting media after the first subculture, the second subculture and the third subculture, respectively (from left to right). (**B**). Matured embryogenic calli and development of somatic embryogenesis under darkness after 8 weeks, 12 weeks and 16 weeks, respectively (from left to right). (**C**). Germinated somatic embryos and shoot multiplication after 4 weeks, 6 weeks and 12 weeks, respectively (from left to right). (**D**). Shoot rooting. (**E**). Acclimatized plantlets. Scale bar: 1 cm.

**Figure 4 plants-12-00644-f004:**
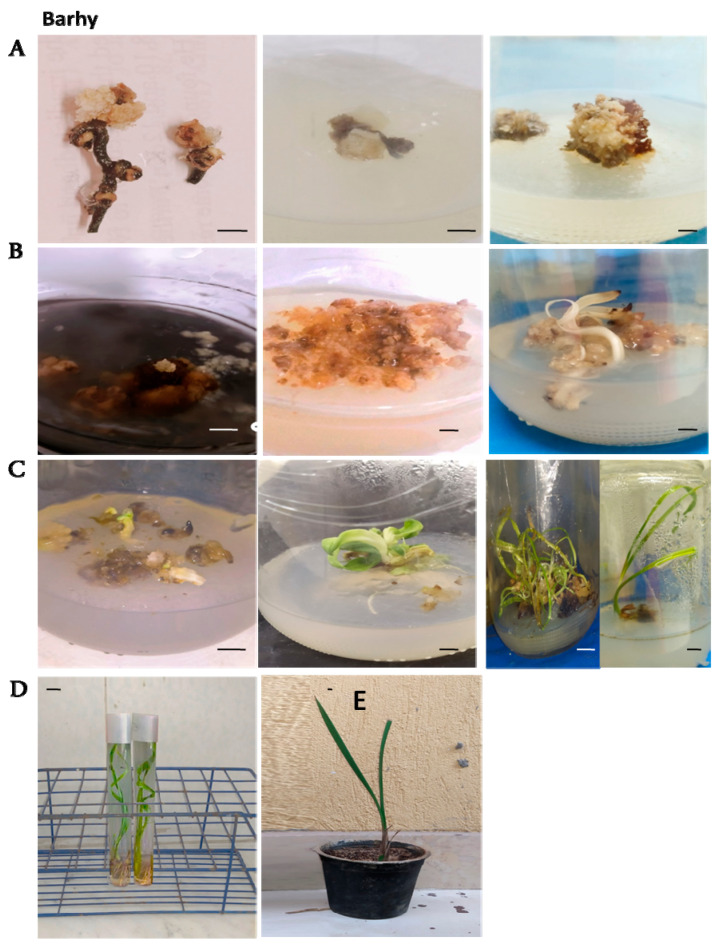
Date palm micropropagation by immature female inflorescences in Barhy variety. (**A**). Various structures of callus formation and development on starting media after the first subculture, the second subculture and the third subculture, respectively (from left to right). (**B**). Matured embryogenic calli and development of somatic embryogenesis under darkness after 8 weeks, 12 weeks and 16 weeks, respectively (from left to right). (**C**). Germinated somatic embryos and shoot multiplication after 4 weeks, 6 weeks and 12 weeks, respectively (from left to right). (**D**). Shoot rooting. (**E**). Acclimatized plantlets. Scale bar: 1 cm.

**Figure 5 plants-12-00644-f005:**
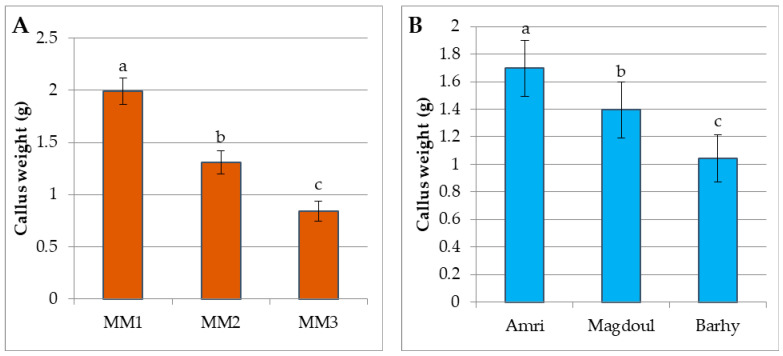
(**A**). The callus weights of the three maturation media (MM1, MM2 and MM3). (**B**). The callus weights of the three varieties (Amri, Magdoul and Barhy). Note: Charts with the same letter are not statistically different at *p <* 0.05. Different letters denote a significant difference among culture media at *p <* 0.05.

**Figure 6 plants-12-00644-f006:**
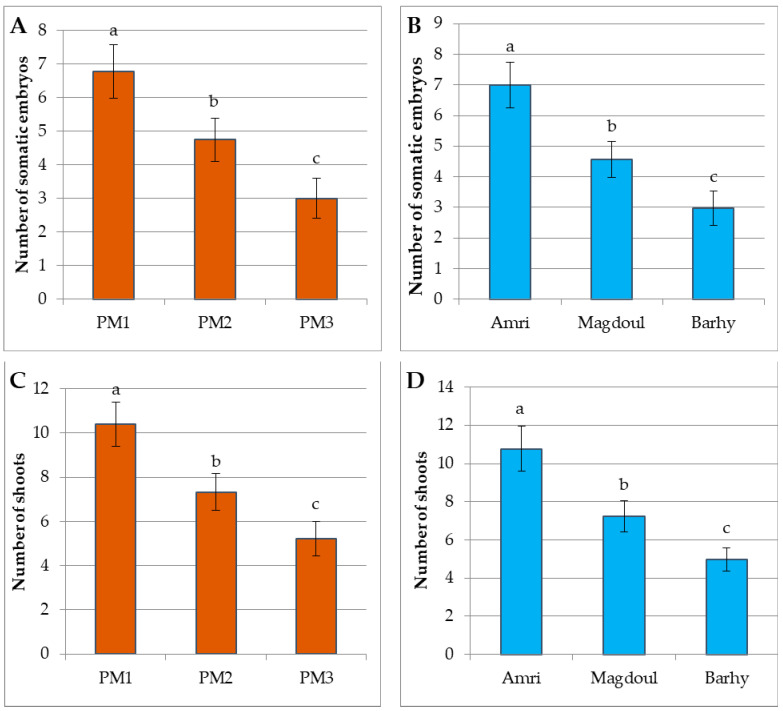
Responses of Amri, Magdoul and Barhy varieties in the multiplication stage. (**A**). Number of somatic embryos of the three multiplication media (PM1, PM2 and PM3). (**B**). Number of somatic embryos of the three varieties (Amri, Magdoul and Barhy). (**C**). The number of shoots of the three multiplication media (PM1, PM2 and PM3). (**D**). The number of shoots of the three varieties (Amri, Magdoul and Barhy). Note: Charts with the same letter are not statistically different at *p <* 0.05. Different letters denote a significant difference among culture media at *p <* 0.05.

**Figure 7 plants-12-00644-f007:**
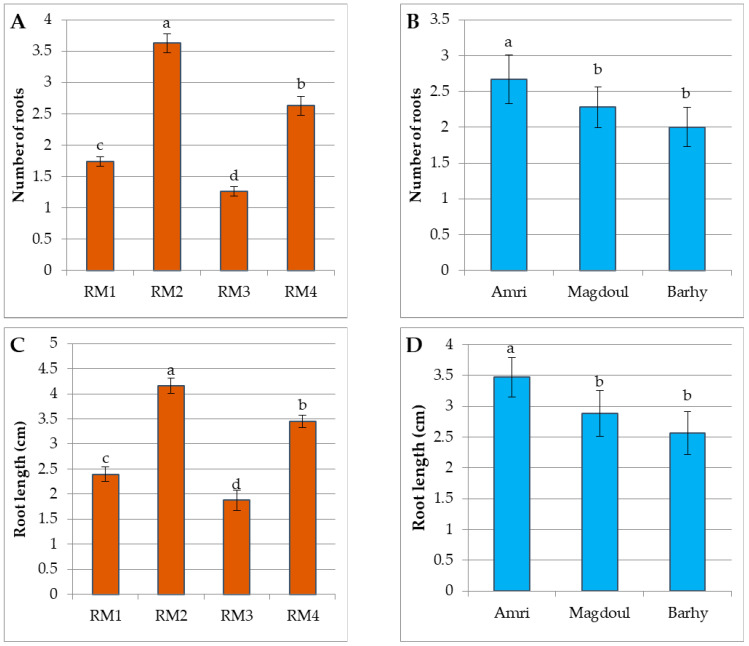
(**A**). Roots numbers of the four rooting media (RM1, RM2, RM3 and RM4). (**B**). Roots numbers of the three varieties (Amri, Magdoul and Barhy). (**C**). The root length of the four rooting media (RM1, RM2, RM3 and RM4). (**D**). The root length of the three varieties (Amri, Magdoul and Barhy). (**E**). The shoot length of the three varieties (Amri, Magdoul and Barhy) on the four rooting media (RM1, RM2, RM3 and RM4). Note: Charts with the same letter are not statistically different at *p <* 0.05. Different letters denote a significant difference among culture media at *p <* 0.05.

**Figure 8 plants-12-00644-f008:**
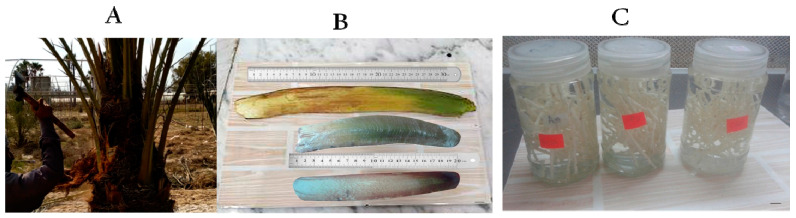
Preparation of immature female inflorescence tissues for date palm micropropagation: (**A**) Spathe harvesting during flowering period. (**B**) Sterilized spathes ready to be opened under aseptic conditions. (**C**) Immature female inflorescence disinfection in sodium hypochlorite solution. Scale bar: 1 cm.

**Table 1 plants-12-00644-t001:** Mean sum of squares (MS) for callus induction frequencies (%), callus weight (g) and number of days to initiate callus in Amri, Barhy and Magdoul varieties.

SOV	Df	MS		
		Callus Induction Frequencies (%)	Number of Days to Initiate Callus	Callus Weight (g)
Replicates	2	75.51	3.595	0.09834
Media	2	4,672.38 **	177.099 **	8.97575 **
Variety	2	5,645.85 **	401.173 **	2.86402 **
Media x variety	4	167.91	12.008	0.02743
Error	70	127.71	10.437	0.28367
Total	80	10,689.36	604.312	12.249
h^2^_b_		82.96%	81.64%	47.68%

Df is degree of freedom, h^2^_b_ is heritability in broad sense, ** indicates *p*-value < 0.01.

**Table 2 plants-12-00644-t002:** Mean sum of squares (MS) of the number of somatic embryos and the shoot number in Amri, Barhy and Magdoul varieties.

SOV	Df	MS	
		Number of Somatic Embryos	Number of Shoots
Replicates	2	14.848	12.913
Media	2	95.623 **	180.803 **
Variety	2	110.612 **	228.831 **
Media x variety	4	0.785	7.041
Error	70	11.281	18.844
Total	80	233.149	448.432
h^2^_b_		43.569%	52.36%

Df is degree of freedom, h^2^_b_ is heritability in broad sense, ** indicates *p*-value < 0.01.

**Table 3 plants-12-00644-t003:** Mean sum of squares (MS) of roots number, root length and shoot length in Amri, Barhy and Magdoul varieties.

SOV	Df	MS		
		Root Number	Root Length (cm)	Shoot Length (cm)
Replicates	2	0.6204	1.9373	0.1548
Media	3	29.4444 **	28.5174 **	81.4038 **
Variety	2	4.0370 *	7.6345 **	15.2823 **
Media x Variety	6	0.1852	0.1444	1.0430 *
Error	94	0.7291	0.3545	0.3796
Total	107	35.016	38.588	98.263
h^2^_b_		22.004%	68.41%	80.11%

Df is degree of freedom: * indicates *p*-value < 0.05: ** indicates *p*-value < 0.01.

**Table 4 plants-12-00644-t004:** Nutrient MS media composition used in several stages of micropropagation in date palm using immature female inflorescences.

Medium Name	Medium Composition (1 L)
Starting media	
SM1	4.4 g MS + 30 g sucrose + 7 g agar + 9 µM 2,4-D + 5.7 µM IAA + 10 µM NAA
SM2	4.4 g MS + 30 g sucrose + 7 g agar + 4.5 µM 2,4-D + 2.85 µM I AA + 5 µM NAA
SM3	4.4 g MS + 30 g sucrose + 7 g agar + 5.7 µM IAA + 5 µM NAA
Maturation media	
MM1	4.4 g MS + 30 g sucrose + 7 g agar + 4.5 µM 2,4-D + 9.8 µM 2-iP + 1.5 AC
MM2	4.4 g MS + 30 g sucrose + 7 g agar +2.25 µM 2,4-D + 4.9 µM 2-iP + 1.5 AC
MM3	4.4 g MS + 30 g sucrose + 7 g agar + 1.125 µM 2,4-D + 2.45 µM 2-iP + 1.5 AC
Multiplication media	
PM1	4.4 g MS + 30 g sucrose + 7 g agar + 4.4 µM BA + 9.8 µM 2-iP
PM2	4.4 g MS + 30 g sucrose + 7 g agar + 2.2 µM BA + 4.6 µM kinetin
PM3	4.4 g MS + 30 g sucrose + 7 g agar + 2.2 µM BA + 4.9 µM 2-iP
Rooting media	
RM1	3.3 g MS + 30 g sucrose + 6 g agar + 0.5 µM NAA + 1.25 µM IBA + 3 g AC
RM2	3.3 g MS + 50 g sucrose + 6 g agar + 0.5 µM NAA + 1.25 µM IBA + 3 g AC
RM3	3.3 g MS + 30 g sucrose + 6 g agar + 0.5 µM NAA + 0.5 µM IBA + 3 g AC
RM4	3.3 g MS + 50 g sucrose + 6 g agar + 0.5 µM NAA + 0.5 µM IBA + 3 g AC

## Data Availability

The data presented in this study are available upon request from the corresponding authors.

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
