# Peer review of "In Vitro Propagation of Three Date Palm (Phoenix dactylifera L.) Varieties Using Immature Female Inflorescences"

_plants, 2023, doi:10.3390/plants12030644_

Round 1

Reviewer 1 Report

Dear authors,

You have done a nice work, however I have found serious flaws in the way results are presented. 

I suggest to revise carefully all the document before submitting it, there are many non-scientific errors throughout the text that make reading the text difficult.

Best regards.

Author Response

Response to Reviewer 1 Comments

Manuscript ID: plants-2062716

Manuscript Title: In vitro micropropagation of three date palm (Phoenix dactylifera L.) varieties using immature female inflorescences

We are very glad that Reviewer 1 highly evaluated our manuscript, and provided constructive comments and valuable suggestions that have helped us further improvement of the quality of our manuscript. All the changes made in response to the Reviewer 1 comments were corrected as track changes in the revised manuscript. We have addressed all of your queries and improved our manuscript following your suggestions as you can see in our point-to-point responses to your comments below.

The manuscript has undergone English language editing by MDPI. The text has been checked for correct use of grammar and common technical terms.

Comments and Suggestions for Authors

You have done a nice work, however I have found serious flaws in the way results are presented. 

I suggest to revise carefully all the document before submitting it, there are many non-scientific errors throughout the text that make reading the text difficult.

Best regards.

Response: Thanks for the reviewer for his/her recommendation.

Abstract

Here and throughout the text please use molarities (µM instead mg/l).

Response: Done accordingly

 Line 19: “Four types of MS media were used” what do you mean by that?

Response: This mean Four types of MS media with four different combinations of growth regulators

Lines 22-23: I would simplify this sentence, for example like this “The highest percentage of callus induction in all the varieties studied was obtained on the starting medium SM1 (2 mg/l 2, 4-D + 1 mg/l IAA+ 5.0 mg/l NAA)”.

Response: Done accordingly. Thank for the reviewer for his proposal.

Lines 24 -28: Please take the name of the medium out of the parenthesis (MM1, PM1, RM2 and RM3). The media don´t achieve of produce anything, the cultures in those media do it, this  make the sentences a bit longer but in my opinion it is the correct way of writing the results.

Examples can be written in another way I put them in case you find them helpful:

Response: Done accordingly. Thank for the reviewer for his proposal.

Line 24: “Cultures in maturation medium MM1 (1 mg/l 2, 4-D + 2 mg/l 2 24 iP + 1.5 g/l AC) achieved the best value. Line 25: “After culture in multiplication medium PM1 (1 mg/l BA+ 25 2 mg/l 2 iP) the highest number of somatic embryos and shoots number were obtained” Line 26: “Explants  in rooting medium RM1 ((RM2: 0.1 mg/l NAA+ 0.25 mg/l IBA + 3 g/l AC), showed…” Line 28: “while the highest shoot length achieved by…” should be “while the highest shoot length was achieved in…”

Response: Done accordingly. Thank for the reviewer for those great corrections.

Lines 30-32: “the analysis of variance revealed highly significant variations among varieties and culture media except for a significant difference in roots number in the rooting stage2 I  don´t understand why you put “except”.

Response: Done accordingly (Corrected)

Line 32: Replace “insignificant” for “non significant”. Line 33: Replace “except in” for “except for”.

Response: Done accordingly (Corrected)

Lines 34-36: If you put the units for the parameters studied, you have to put it all of them not only for frequencies and weight, instead I suggest to delete them from the abstract (% and g).

Response: Done accordingly. Thank for the reviewer for those great corrections.

Lines 37-38: i would change could for can in both cases.

Response: Done accordingly. Thank for the reviewer for those great corrections.

Keywords: Put the keywords in alphabetical order, please.

Response: Done accordingly (Corrected)

Abbreviations: The first time an abbreviation appears the full name of it should be given (plant growths regulators, or media for example). I would also include an abbreviations section.

Response: Done accordingly

Introduction

Line 59: Replace “palnts” for “plants”

Response: Done accordingly

Line 61: the shoot tip doesn´t encounter issues, when culturing shoot tips issues are encountered. Then, rewrite the sentence, please.

Response: Done accordingly

Lines 62-64: “The inflorescences were easily converted to a vegetative state, therefore, the use of meristematic floral spikes at an early stage aims to achieve this purpose” I don´t understand this sentence. What do you mean by were easily converted to a vegetative state? Explain, please.

Response: this sentence mean that the use of inflorescences as explant is easy to differentiate to vegetative organs, so they must be in meristematic stage to be more suitable as explants.

Line 65: “this early-stage changes depending on the variety, …” rewrite “these early-stage  changes depend on the variety,…”

Response: Done accordingly

Line 66: “the ability of the floral tissue to reverse into a vegetative state” when you say vegetative state, are you talking about the responsiveness of this type of explant?

Response: Yes, this means that the type of the floral tissues (reproductive tissues) converted to vegetative tissues.

Line 68: “growth hormones” I would better say plant growth regulators.

Response: Done accordingly

Lines 69-71: What is the light shed here? I mean, you want to point out what it is novel in your  study, in this case I supose the initial explant, but that doesn´t shed light about the process itself. Rewrite, please.

Response: Done accordingly

Lines 71-73: “Different hormone combinations were used to accomplish the optimal response during the different stages of micropropagation” this is a conclusión, I think it shouldn´t appear in the introduction.

Response: Done accordingly

Line 78: Determine not determin.

Response: Done accordingly

Results:

Line 84: “different plant growth regulators” instead of “various plant hormones”.

Response: Done accordingly

Figure 1 and Figure 5 and Figure 6: According to table 1 and table 2 the interaction medium x variety was not significant for callus induction frequency %, number of days, callus weight, number of somatic embryos and number of shoots. Then you cannot perform a post hoc test as if there was a significant interaction. In fact other statistics programs as SPSS do not offer this possibility. If there are not significant differences you could have a figure comparing media and another one comparing varieties (with letter indicating which one is different from the other). Of course, the results in the text should be explained accordingly.

Response: Done accordingly

Lines 106-107: “The weight of callus and embryogenic callus was strongly influenced by …” How do you differientiate callus and embryogenic callus?

Response: The non-embryogenic callus has characteristics of compact structure, white or milky white, smooth, and nontransparent and tends to blackish brown color. -The early stages of embryo formation begin with the formation of mass pro embryo mass (PEM) or the pro globular embryogenic callus which has the structure of a glossy, transparent, and dried.

Line 107 and line 112: 2 ip and 2iP are the same? Please be consistent with the abbreviation and revise them throughout the text; and, as said before, give the full name of the the first time they´re mentioned.

Response: Done accordingly

Line 111-112: The medium does not possess anything, the callus grew there might have  exhibit a better growth, this is only an example of expressions that should be changed throughout the text.

Response: Done accordingly

Line 114: “%” between parenthesis.

Response: Done accordingly

Figure 7A and 7B: As mentioned for figures 1,5 and 6; according to table 3 the interaction medium x variety was not significant for number of roots and root lenght. Then you cannot performed a post hoc test as if there was a significant interaction. If there are not significant differences you could have a figure comparing media and another one comparing varieties (with letter indicating which one is different from the other). Of course, the results in the text should be explained accordingly. Figure 7C is correct.

Response: Done accordingly

Lines 194-197: “Pre-acclimatization is an in vitro hardening … to produce healthy and vigorous growing plantlets for acclimatization”. These sentences are not results, shoud be removed from this section.

Response: Done accordingly

Lines 200-201: “The percentage of acclimatized plantlets was calculated by dividing the number of 200 plants that successfully adapted to their new environment by the total number of plants” this description should be in Material and Methods section.

Response: Done accordingly

Discussion:

I cannot revise this section until the previous ones have been revised carefully and rewritten.

Response: Done accordingly

In any case, I suggest to revise it according to the changes proposed for results and for the English style.

Response: Done accordingly

Materials and methods:

Lines 277-278: Plastic tissue? Do you mean plastic film? If so, change it, please.

Response: Done accordingly

Line 280: “Sphates” instead of “Spahtes”.

Response: Done accordingly (corrected to spathes)

Line 283: “spikelet” without capital letter.

Lines 286-287: As an abreviation for litre I would use L instead of l throughout the text.

Response: Done accordingly

 Line 288: “were” instead of “have been”

Response: Done accordingly

Line 290: Which agar type or brand did you use? Specify it in the text.

Response: Done accordingly (PTC agar, sigma)

Line 292: What type of jars did you use, volume or brand? Specify it in the text.

Response: Done accordingly

Line 295-296: Repetitive, you have mentioned something similar in Plant materials.

Response: The method here in more details

Line 296: I think you are not referring to table 1 (table 4?). The same in line 305.

Response: Done accordingly

Line 308: How did you measure callus weight?

Response: The callus weight was measured at the end of the subculture as the jar weight with calli minus the jar weight without calli under sterile conditions (the reference was added in 4.5. Measurements section). 

Line 312: when calculating the number of embryos/jar, did you measure the tissue you were  placing in each jar? and the same for the number of shoots/jar.

Response:  The explants placed in jars were cut into parts of 3 cm length including 3-4 immature florets. The same was done with number of shoots per explant.

Line 319: Delete shoots.

Response: Done accordingly

Line 320: “plus” instead “+”; “activated” instead “Activated”.

Response: Done accordingly

Line 321: If AC stands for activated charcoal, specify it inside parenthesis in line 320.

Response: Done accordingly

 Lines 322-324: Instead of present tense, use past tense as in the rest of the text.

Response: Done accordingly

Line 326: “9000” instead of “9,000”.

Response: Done accordingly

Line 329: Which was the fungicide solution? Specify, please.

Response: Done accordingly. Rezolex Fungicide for Plants (100g)

Line 330: “mixture” instead “mixtures”.

Response: Done accordingly

Line 335: “%” instead “percent”.

Response: Done accordingly

Line 340: You don´t use the CRD abbreviation anymore, so I suggest to eliminate it.

Response: Done accordingly

Line 341: “with” instead “was”. What do you mean by elongation stage, or it goes together with rooting stage?

Response: Done accordingly. Yes, corrected

Line 342: “Each treatment consisted of three replications” what was each replication, one jar?  If so haw many explants did you put in each jar? Please explain better this part.

Response: Done accordingly (4.4. Culture conditions section )

Line 345: LSD, do you mean the post-hoc test of Fisher? If so you should rewrite this part, saying something like when significant differences were found, or when necessary…LSD post-hoc was used for multiple comparisons.

Response: Done accordingly (Many thanks for this proposal)

Table 4: The tables should be self standing, therefore you should explain in the Table capture all the abbreviations used in the table.

Response: Done accordingly. All the abbreviations were listed in abbreviations section you proposed. (Many thanks).

Reviewer 2 Report

The manuscript might be an interesting research, however it needs major changes. 

First of all, the authors should have the manusript corrected by a professional or native speaker, familiar with academic writing. The language is quite complicated to read and there are also grammar mistakes and incorrect subject forms. 

The other thing is presentation of the results. While reading the manuscript, one has to search for the results. They should be presented close to the writing. It is also difficult to find the values which are described. FIgure 1 and Table 1 are described in the beginning, while Figure 2 is presented before Table 1. I cannot see the connection between callus frequency and number of days for induction. I would recommend to change the presentation of the results, to make it easier. The results described in one paragraph could be presented together, in a table for example. 

Authors should also change the discussion to present the importance of the study. It seems there are many studies on date palm micropropagation. 

The M&M: what was the replication? how many individuals were used per repetition? What kind of light was used? 

Statistical analysis: was the normal distribution checked? 

Some more information is given in the manuscript, but I have suggested corrections only in few places. 

Author Response

Response to Reviewer 2 Comments

Manuscript ID: plants-2062716

Manuscript Title: In vitro micropropagation of three date palm (Phoenix dactylifera L.) varieties using immature female inflorescences

We are very glad and thankful for the great scientist (Reviewer 2) evaluating of our manuscript, and provided constructive comments and valuable suggestions that have helped us further improvement of the quality of our manuscript. All the changes and corrections made in response to the Reviewer 2 comments were corrected as track changes in the revised manuscript. We have addressed all of your queries and improved our manuscript following your suggestions as you can see in our point-to-point responses to your comments below.

The manuscript has undergone English language editing by MDPI. The text has been checked for correct use of grammar and common technical terms.

Comments and Suggestions for Authors

The manuscript might be an interesting research, however it needs major changes. 

First of all, the authors should have the manusript corrected by a professional or native speaker, familiar with academic writing. The language is quite complicated to read and there are also grammar mistakes and incorrect subject forms.

Response: Done accordingly 

The other thing is presentation of the results. While reading the manuscript, one has to search for the results. They should be presented close to the writing. It is also difficult to find the values which are described. Figure 1 and Table 1 are described in the beginning, while Figure 2 is presented before Table 1.

Response: Done accordingly 

I cannot see the connection between callus frequency and number of days for induction. I would recommend to change the presentation of the results, to make it easier. The results described in one paragraph could be presented together, in a table for example. 

Response: the connection between callus frequency and number of days for induction is that the better media give high callus frequency in a short period.

Authors should also change the discussion to present the importance of the study. It seems there are many studies on date palm micropropagation.

Response: Done accordingly 

The M&M: what was the replication? how many individuals were used per repetition? What kind of light was used?

 Response: Done accordingly

Statistical analysis: was the normal distribution checked?

Response: Thanks for the reviewer for this comment. The normal distribution was not applied because we used the analysis of variance and LSD0.05 tests  

Some more information is given in the manuscript, but I have suggested corrections only in few places.

Response: Done accordingly. Many thanks for the reviewer for his/her great efforts and valuable corrections 

Some required explanations

1-      Heritability?

Response: is the ratio of total genetic variance to total phenotypic variance. The sense to calculate it is to determine the traits that could be genetically improved (possess high heritabiltiy) and the traits that could be improved by the nutrient media and plant growth regulators (possess low heritabiltiy)

2-      3.3  g/l  MS?

Response: means three-quarters strength of MS medium (the full strength is 4.4 g/l)

  • The percentage of acclimatized plantlets was 86.67% in the Amri variety followed by Magdoul 82.33% and Barhy 77.56%

Response: This result just by dividing the surviving plants by the total plants introduced for acclimation

  • The third comment in discussion section

Response: this in a citation not a result of the current study

Reviewer 3 Report

I reviewed the paper titled In vitro micropropagation of three date palm (Phoenix dactylifera L.) varieties using immature female inflorescences. The authors aimed to determine the ideal tissue culture media for the success of date palm micropropagation using immature female inflorescences and to provide a new protocol for micropropagation of date palm without sacrifying the adult trees or the offshoots. The results are interesting and applicable. But the authors should address the following questions:

The manuscript has several typos and the English is poor. Therefore, the paper must be edited by a native language editor.

Introduction and discussion are short. It is recommended to enrich them. For example it is recommended to address the following papers:

Vahdati K, Jariteh M, Niknam V, Mirmasoumi M and Ebrahimzadeh H (2006) Somatic embryogenesis and embryo maturation in Persian walnut. Acta Horticulturae. 705:199-205

Hatami, A., Abootalebi Jahromi, A., Ejraei, A., Mohammadi Jahromi, A. H., & Hassanzadeh Khankahdani, H. (2023). Study of Biochemical Traits and Mineral Elements in Date Palm Fruits under Preharvest Foliar Application of Organic Fertilizers and Micronutrients. International Journal of Horticultural Science and Technology10(3), 125-140. 

Asayesh ZM, Vahdati K, Aliniaeifard S (2017) Investigation of physiological components involved in low water conservation capacity of in vitro walnut plants. Scientia Horticulturae. 224: 1-7.

In line 24, the authors mentioned, these results showed that the MM1 medium (1 mg/l 2, 4-D + 2 mg/l 2iP + 1.5 g/l AC) possessed the best performance for callus weight. What is the role of activated charcoal in the maturity stage?

In general, we use BAP as cytokinin, why was it preferred to use 2ip in this article?

Why were 3 types of auxin used in the starting stage?

Does the use of 2,4 D have no effect on somaclonal variation?

In line 69, it is mentioned that RM2 medium (3.3 g/l MS +  50 g/l sucrose + 6 g/l agar + 0.1 mg/l NAA + 0.25 mg/l IBA + 3 g/l AC). Why in rooting medium, you increased the sugar?

Please add some sentences about applicability of the results in commercial tissue culture labs, if any.

Author Response

Response to Reviewer 3 Comments

Manuscript ID: plants-2062716

Manuscript Title: In vitro micropropagation of three date palm (Phoenix dactylifera L.) varieties using immature female inflorescences

We are very glad that Reviewer 3 highly evaluated our manuscript, and provided constructive comments and valuable suggestions that have helped us further improvement of the quality of our manuscript. All the changes made in response to the Reviewer 3 comments were corrected as track changes in the revised manuscript. We have addressed all of your queries and improved our manuscript following your suggestions as you can see in our point-to-point responses to your comments below.

The manuscript has undergone English language editing by MDPI. The text has been checked for correct use of grammar and common technical terms.

Comments and Suggestions for Authors

I reviewed the paper titled In vitro micropropagation of three date palm (Phoenix dactylifera L.) varieties using immature female inflorescences. The authors aimed to determine the ideal tissue culture media for the success of date palm micropropagation using immature female inflorescences and to provide a new protocol for micropropagation of date palm without sacrifying the adult trees or the offshoots. The results are interesting and applicable. But the authors should address the following questions:

The manuscript has several typos and the English is poor. Therefore, the paper must be edited by a native language editor.

Response: Done accordingly 

Introduction and discussion are short. It is recommended to enrich them. For example it is recommended to address the following papers:

Vahdati K, Jariteh M, Niknam V, Mirmasoumi M and Ebrahimzadeh H (2006) Somatic embryogenesis and embryo maturation in Persian walnut. Acta Horticulturae. 705:199-205

Response: Done accordingly 

Hatami, A., Abootalebi Jahromi, A., Ejraei, A., Mohammadi Jahromi, A. H., & Hassanzadeh Khankahdani, H. (2023). Study of Biochemical Traits and Mineral Elements in Date Palm Fruits under Preharvest Foliar Application of Organic Fertilizers and Micronutrients. International Journal of Horticultural Science and Technology, 10(3), 125-140.

Response: Done accordingly  

Asayesh ZM, Vahdati K, Aliniaeifard S (2017) Investigation of physiological components involved in low water conservation capacity of in vitro walnut plants. Scientia Horticulturae. 224: 1-7.

Response: Done accordingly 

In line 24, the authors mentioned, these results showed that the MM1 medium (1 mg/l 2, 4-D + 2 mg/l 2iP + 1.5 g/l AC) possessed the best performance for callus weight. What is the role of activated charcoal in the maturity stage?

Response: The activated charcoal improves cell growth and development. The activated charcoal promotes the morphogenesis due to its irreversible adsorption of inhibitory compounds in the culture medium and substancially decreasing the toxic metabolites, phenolic exudation and brown exudate accumulation.

In general, we use BAP as cytokinin, why was it preferred to use 2ip in this article?

Response: As mentioned in discussion section and based on previous studies that 2-ip increase the proliferation rate and cell differentiation.

Why were 3 types of auxin used in the starting stage?

Response: This is a new trial to promote the induction of callus and decrease the number of days required for callus induction.

Does the use of 2,4 D have no effect on somaclonal variation?

Response: In discussion section, several studies reported that 2, 4-D considered one of the most helpful auxins being utilized in effective successful cultures. Several studies documented the beneficial effects of various treatments of 2, 4-D in combination with other growth regulators to promote the embryogenesis of calli in various date palm varieties. All growth regulators may cause somaclonal variation when used in high concentrations with the aid of other factors like number of subcultures.

In line 69, it is mentioned that RM2 medium (3.3 g/l MS +  50 g/l sucrose + 6 g/l agar + 0.1 mg/l NAA + 0.25 mg/l IBA + 3 g/l AC). Why in rooting medium, you increased the sugar?

Response: The root initiation and plant growth need high energy and it can be obtained using metabolic substrates as carbohydrates (Thorpe, 1982).The sucrose acts as a source of energy and plays an important role in maintaining osmoticum (Cuenca &Vieitez, 2000). Increasing the concentration of sucrose from 30-60 g/L led to a gradual increase in the number of roots and roots length.

Please add some sentences about applicability of the results in commercial tissue culture labs, if any.

Response: Done accordingly 

Round 2

Reviewer 1 Report

Dear authors,

Enclosed you can find my comments, i still find not adequate the interpretation and representation of the results of the statistical analyses. I hope my comments will help you.

Best regards.

Author Response

Response to Reviewer 1 Comments (Round 2)

Manuscript ID: plants-2062716

Manuscript Title: In vitro micropropagation of three date palm (Phoenix dactylifera L.) varieties using immature female inflorescences

We are very glad that Reviewer 1 highly evaluated our manuscript, and provided constructive comments and valuable suggestions that have helped us further improvement of the quality of our manuscript. All the changes made in response to the Reviewer 1 comments were corrected as track changes in the revised manuscript. We have addressed all of your queries and improved our manuscript following your suggestions as you can see in revised version of the manuscript. Figures for all the parameters analyzed comparing the media (independently of the variety), and another figures comparing the varieties independently of the media were listed in the manuscript. The text was rewritten accordingly. Many thanks for your care about the quality of the manuscript.

Reviewer 2 Report

I would like to thank the Authors for all the changes done. I still have just few remarks, given in the manuscript.

Author Response

Response to Reviewer 2 Comments (Round 2)

Manuscript ID: plants-2062716

Manuscript Title: In vitro micropropagation of three date palm (Phoenix dactylifera L.) varieties using immature female inflorescences

We are very glad that Reviewer 2 highly evaluated our manuscript, and provided constructive comments and valuable suggestions that have helped us further improvement of the quality of our manuscript. All the changes made in response to the Reviewer 2 comments were corrected as track changes in the revised manuscript. We have addressed all of your queries and improved our manuscript following your suggestions as you can see in revised version of the manuscript. Many thanks for your care about the quality of the manuscript.